# Force Sensation Induced by Electrical Stimulation of the Tendon of Biceps Muscle

**Akifumi Takahashi** [1,2,*] **and Hiroyuki Kajimoto** [1]

1   Department of Informatics, The University of Electro-Communications, 1-5-1 Chofugaoka, Chofu, Tokyo 182-8585, Japan; kajimoto@kaji-lab.jp
2   Research Fellow of the Japan Society for the Promotion of Science, 5-3-1 Kojimachi, Chiyoda-ku, Tokyo 102-0083, Japan
*   Correspondence: a.takahashi@kaji-lab.jp; Tel.: +81-42-443-5445

**Abstract:** Many wearable interfaces have been proposed to present force to the upper limb and elbow joint. One way to achieve a compact wearable haptic interface is to use electrical stimulation, and we have suggested that transcutaneous electrical stimulation above the wrist tendon can produce force a sensation in the direction of the muscle stretching; however, it has not been investigated in detail whether the force sensation presented by the electrical stimulation of the tendon occurs in the upper limb joints. In this study, to investigate whether the force sensation is generated when applying electrical stimulation of the skin at the tendon or at the muscle belly of the biceps brachii muscle, we quantitatively evaluated the direction and amount of the force sensation under the aforementioned conditions. The results showed that the electrical stimulation of the tendon produced significant force sensation in the direction of elbow extension. On the other hand, in some participants, the electrical stimulation of the muscle belly worked as a supporting force, resulting in the sensation of weakened force perception. In general, we concluded that the sensation produced by muscle stimulation was different from that produced by stimulation of the tendon.

**Keywords:** haptics; sense of force; proprioception; tendon; electrical stimulation; biceps; Golgi tendon organ

## 1. Introduction

When a person supports an object, a force interacts between the object and the body. This force causes localized skin deformation and provides detailed shape and texture information. Additionally, a certain amount of force attempts to bend limbs around a joint, changing muscle length if the limbs are physically bent or making the motor nerves excited and then causing the muscles to contract if a person tries to resist the external force. These haptic sensations provide detailed information about the interaction between the body and objects that cannot be obtained through vision alone. In particular, movements around the joints are perceived by the receptors around the muscles. Such a haptic sensation is called proprioception.

Many methods of artificially presenting proprioception have been proposed for remote control and work in VR spaces. Many of these methods have used electromagnetic motors to physically present force, making it possible to present precise force sensation and even skin sensation. However, the multiple degrees of freedom of motion require the same number of motors, causing an increase in the size of the device in order to have multiple degrees of freedom [1,2]. Indeed, some studies showed that one actuator could cover a multi-dimensional haptic feedback motor system [3,4]; however, the situation where the technique is available is limited, and a heavy motor is still required for a large degree of torque. On the other hand, a method to present force by controlling human muscles with electrical stimulation (EMS) has been proposed [5–11]. This method requires only a tiny electrical circuit board for power supply, one for stimulus control, and electrodes to be

placed on the human body, the net size of all of which is much smaller than the devices using electromagnetic motors. This method can be regarded as a method that uses motors in vivo (i.e., muscles) instead of the external electromagnetic motors, which is the main reason for the large size. On the other hand, EMS inevitably involves muscle contraction, which does not usually occur in natural situations, and thus also produces sensations that are different from the actual sensations.

EMS stimulates motor nerves and therefore requires actual movement to produce sensation. On the other hand, the authors have proposed a method of presenting force sensation by stimulating sensory nerves related to proprioception. We especially focused on Golgi tendon organs, which are related to muscle tension and force. We targeted their nerves called Ib fiber from electrodes on the skin surface as a method that does not require the generation of actual movement [12–14]. In the previous report, we stimulated the dorsal wrist's tendon connected to the muscle that extends the wrist. We speculated that if we could stimulate the Golgi tendon organ, we could present the information that this muscle is tensed, while the muscle length does not change because no movement is physically generated. Additionally, such an electrical stimulation of afferents can ignore the mechanical structure of the mechanoreceptor to a certain extent; thus, this method has the potential to render haptics, keeping the footprint small even as EMS. As a result of the experiment, we confirmed it is possible to present a force of about 300 gf ($\simeq$ 2.94 N) [12], and the effect was enhanced by multimodal stimulation with visual and cutaneous sensation [13]. Here, [gf] stands for the unit of gram force and $1.0\ \mathrm{gf} = 1.0 \times 10^{-3}\ \mathrm{kgf} = 9.8 \times 10^{-3}\ \mathrm{N}$.

However, it has not been investigated in detail whether this method also works in other upper limb joints. The muscles in the upper arm are mostly used for lifting and supporting objects, and many wearable interfaces have presented force to the upper limb and elbow joint [10,11,15,16].

This study investigates whether the force sensation in extending the elbow joint is generated by tendon part stimulation and compares it with the conditions of stimulating biceps brachii muscles.

## 2. Related Work

### 2.1. Proprioception

One of the senses related to movement comes from motor command [17,18]. The efferent signal from the central nervous system that causes muscle contraction makes a corollary sensation. However, there have been cases in which the afferent nerves failed to function, and the loss of information from the peripheral sensors made it difficult to move the body freely, even though the efferent nerves were not paralyzed [19]. Therefore, proprioception cannot be established only by such a forward model using the coronary sensation from the motor signal in the central. Feedback from the peripheral is also essential, which is obtained mainly by mechanoreceptors around skeletal muscles; moreover, it is also integrated with cutaneous sensation and vision.

There are two main proprioceptors around skeletal muscles: one is the muscle spindle, and the other is the Golgi tendon organ. The muscle spindle can obtain information about the length of the muscle and its changes [17,18], and the Golgi tendon organ can obtain information about the force with which the muscle contracts while having a high threshold for stimuli that change the length of the muscle [20].

### 2.2. Stimulation of Tendons and Their Effects

2.2.1. Tendon Stimulation Focusing on Muscle Spindles and Observation of Reflexes and Motor Illusions

Goodwin et al. [21] observed that the tendons on the elbow side of the biceps and triceps muscles produced the illusion of motion when subjecting them to vibratory stimulation. In their experiment, a vibrating stimulus was applied to one arm of the participants, and they were instructed to reproduce the motion of the vibrating arm with the other arm. They confirmed the illusion of "motion" in the direction in which the vibrated muscle was

stretched (e.g., the vibration of the biceps tendon makes the sensation as if the forearm was extended). The intensity of the motion illusion varied with the intensity of the vibration stimulus; in particular, a clear correlation was observed in the vibration frequency, suggesting that around 80 Hz is optimal [22]. Tonic vibration reflex (TVR), which weakens the efferent signal to the antagonist muscle, is also known to occur when the intensity of tendon vibration exceeds a certain level [23].

These methods, which provide the sense of movement without actually moving the body, can make clinical applications for rehabilitation or VR applications. Conrad et al. [24] confirmed that Ia nerve stimulation by tendon vibration improved motor performance in stroke patients. Several attempts have been made to improve the operability of an arm in VR space [25] or a robotic prosthetic hand [26] when operating a BCI. In addition, several augmented reality applications have attempted to modulate real body motion [27]. Furthermore, Roll et al. showed the possibility of presenting kinematic sensations in arbitrary two or three dimensions [28].

In addition, Gandevia et al. [29] reported that electrical stimulation from the skin surface electrodes aiming at the median nerve in the wrist, including Ia fiber, produced a similar kinesthetic sensation to needle electrodes. In addition, during EMS, not only the alpha fiber but also the Ia fiber can be stimulated; especially when stimulating the tibia, a muscle contraction caused by alpha nerve stimulation (M wave) was observed, followed by the one that seemed to be a reflex caused by Ia nerve stimulation (H wave) [30]. In addition, it has been confirmed that electrical stimulation of the muscle belly produces a sensation similar to a motion illusion [31].

### 2.2.2. Tendon Stimulation Focusing on Golgi Tendon Organs and Observation of Reflex and Force Illusion

Most studies using stimulation of the Golgi tendon organs or Ib nerves have focused on the inhibitory reflex called the Ib reflex. Kahn and colleagues used surface electrodes, confirming that stimulation of the gastrocnemius tendon caused inhibition of muscle contraction [32]. The cathode was placed in the middle of the gastrocnemius muscle (1 cm from the myotendinous junction on the heel side), and the anode was placed next to the cathode so that no muscle stimulation occurred. On the other hand, it is also known that signals from the Ib nerve can cause excitatory reflexes rather than inhibitory reflexes, depending on the situation, since they pass through many intervening nerves. For example, the signal of the Ib nerve in the tibia has a positive feedback effect when feet are hitting the ground [33,34].

The Golgi tendon organ has a high threshold to muscle stretch or vibration stimuli during relaxation [35], while the optimal stimulus to the Golgi tendon organ is muscle contraction [17]. Therefore, if selective stimulation to Golgi tendon organs or Ib fibers is possible, force sensation due to virtual muscle contraction, such as a sensation of supporting a virtual object pushing the arm, can be generated. In our previous work, we reported that electrical stimulation to the wrist extensor's tendon produced a force sensation as if the arm was pushed from the back of the hand [12–14]. As an application to VR, Kaneko et al. [36,37] proposed to present the sensation of stepping forward by electrical stimulation above the tendons of the feet. Yem et al. [38] reported that stimulation of the tendons on the back of the fingers produces a force sensation that makes the fingers bend, and they applied this to the presentation of a stickiness sensation in VR. However, the mechanism of force sensation presentation by such electrical stimulation above tendons has not yet been clarified. If the Golgi tendon organ mainly contributes to the sensations, one explanation is that it directly stimulates the Ib nerve, and another is that it may indirectly stimulate the Golgi tendon organ by slightly contracting the muscle as it does not cause significant arm's movement [35].

### 2.2.3. Sense of Force and Sense of Effort

We have introduced methods to create motion and force illusions by stimulating tendons, and both of these methods have been reported to contribute to "force perception".

However, the sense of force is divided at least into two types. One is the Sense of Force, and the other is the Sense of Effort. The former can be used in various contexts, but it is the sense fed back regarding the force applied to the end of the body in contrast to the latter, and the latter is the sense of feedforward, i.e., the sense derived from the command to apply force. The latter is a feedforward sensation, i.e., a command-derived sensation. However, the sense of force is sometimes used naively as an integrated sensation that includes the sense of effort, and the sense of effort has been reported as it could be adjusted in the upper layers of the motor cortex by feedback information from the peripheral nerves [39]. In addition, it has been reported that stimulation of muscle spindles, which induce motor illusions, also contributes to the sense of force [40]. It is possible that people can actually perceive these sensations separately. For example, in an experiment in which a participant was asked to press a spring, it was reported that when the participant was asked about the effort of pressing the spring while a muscle relaxant was injected, the participant felt they had to exert more force than usual, whereas when the participant was asked about the stiffness of the spring, the result was the same as in normal conditions [41].

In this study, we focused on the sense of force as the information from the limb, and the instructions to the participants were based on this.

## 3. Materials and Methods

The main purpose of this study was to determine whether electrical stimulation over the biceps' tendon also produces a force sensation. We also would like to verify the mechanism by which this force sensation is generated. The study was conducted according to the guidelines of the Declaration of Helsinki and approved by the Institutional Review Board of The University of Electro-Communications (#18044).

### 3.1. The Hypothesis of the Experiment and Experimental Conditions

In this experiment, electrodes were placed above the tendon of the biceps brachii muscle (Figure 1a) aiming at the Golgi tendon organ and its nerve, Ib fibers, at the musculotendinous junction, so that the stimulation could give a user the impression that the biceps were contracting more strongly than they actually were. As the users also found that the angle of the limb did not change (because real force does not change), they could feel virtual external force opposing the virtual contraction. When seeking the muscle belly and tendon of a participant, we asked the participant to have and keep the same posture illustrated as Figure 1b and pulled the participant's forearm in the extending direction at a certain intensity to make the muscle and tendon stand out. Then, we considered the thickest part as the muscle belly, the part where the muscle gets gradually thinner as the musculotendinous junction, and the thin part as the tendon.

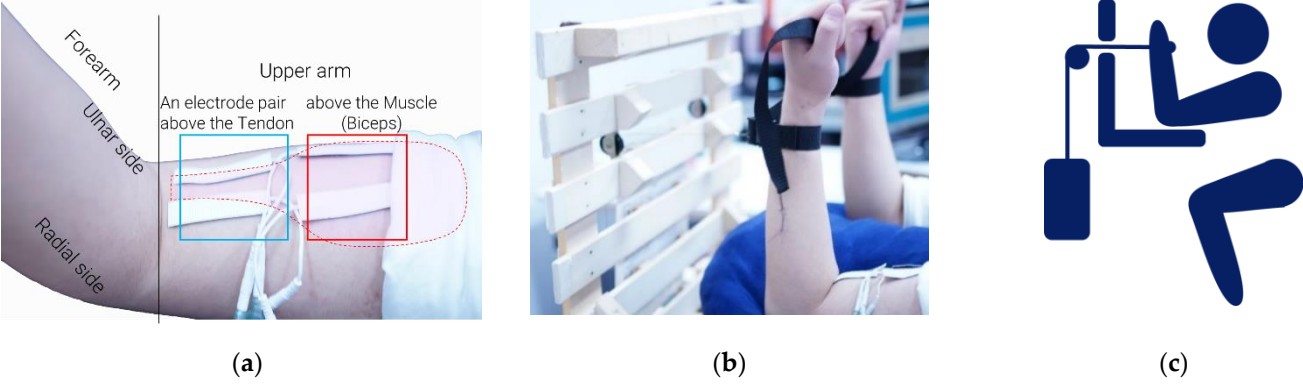

|  |  |  |
|:--:|:--:|:--:|
| (**a**) | (**b**) | (**c**) |

**Figure 1.** (**a**) Electrodes' position. A pair of electrodes is put above the biceps muscle of the left arm, and the other pair is put over the tendon and the junction between the muscle and tendon. (**b**) Experimental setup for comparison of force sensation. (**c**) A conceptual side view of (**b**).

In the previous report [12–14], we presented only electrical stimulation. This study combined it with actual force presentation generated by weights to investigate how much the force sensation would be enhanced by electrical stimulation.

On the other hand, the stimulus above the tendon might eventually become a muscle stimulus and cause a weak muscle contraction, affecting the Golgi tendon organ and muscle spindles and producing force sensation. However, according to the conventional EMS work [5–11] that stimulated the same muscle to mimic an external force, they produced a force sensation opposite to the one produced by stimulating the tendon organ. In order to investigate whether our proposing method is based on muscle stimulation or not, we placed electrodes not only on the tendon but also on the muscle belly of the biceps muscle (Figure 1a) as a condition that actively induces muscle contraction.

Therefore, we measured and compared the magnitude of the force sensation in each of the three conditions: electrical stimulation of the tendon of the biceps, electrical stimulation of the muscle belly, and no electrical stimulation. The electrodes' positions were decided heuristically, which can be considered reasonable to a certain extent because the effect of electrical stimulation grows weak quickly if an electrode's position is moved from the position no more than 1 cm.

Our hypothesis is as follows. Stimulation of the tendon will increase force sensation, while stimulation of the muscle belly will decrease it because the stimulation will act as a force driving in the direction of elbow flexion (in the opposite direction to the force due to the weight) as in the conventional method [5–11].

The elbow was placed on the desk so that the other proximal joint was not involved in the movement (Figure 1b).

### 3.2. Measuring the Magnitude of Force Sensation

In the experiment, the arm was pulled horizontally by weight through a pulley using the apparatus shown in Figure 1b,c. The force applied to the left arm was fixed at 400 gf ($\simeq 3.92$ N), and the weight applied to the right arm was changed. The weights were adjusted to give the same force sensation as that given to the left arm by PEST [42] adaptive physical experiment method. The participants were not told which weight connecting the right or the left arm would be changed. PEST can be regarded as a type of staircase method, in which the step (stimulus intensity: the weight pulling the right arm) is changed according to the participant's answer as to which weight is heavier. The rule of the change is as follows:

1.  Initial condition: 400 gf traction force on the left arm and stimulation of one of the three conditions on the upper arm. A force of 1200 gf ($\simeq 11.8$ N) is applied to the right arm.
2.  In each step, the stimulus was given four times under the same condition in the same step. If the participant answered that the force sensation on the right/left side was stronger more than two times, the force presented to the right arm was made lighter/heavier according to the rule of 3. When the answer was 2:2, the fifth stimulus was performed in the same step, changing the step according to the answer.
3.  The step size was reduced by half every time the participant's response was reversed (e.g., the participant responded that the force perception on the right side was greater at the n-th step, and the participant responded that the force perception on the left side was greater at the n+1th step). Conversely, when the stimulus increased or decreased in the same direction more than three times, the step size was doubled after the third time. The maximum step size was 400 gf, and the step size was halved if the force presented to the right side would become less than 0 gf ($= 0$ N).
4.  Termination condition: we terminated when the step size reached 25 gf ($\simeq 0.245$ N) and used the result as an estimate.

We ran all the estimation procedures for all the three conditions in parallel, and each trial of the three conditions was conducted randomly in a step, e.g., Figure 2 shows a typical example of one subject. In the experiment, participants were allowed to move their arms

slowly within the range where the strings were considered to be parallel to the ground. The range was instructed as $10° \sim 20°$, but we did not measure or feedback the exact angle. The maximum experimental duration per participant was two and a half hours, and the maximum duration of one trial was less than two minutes. There was a 10~30 s interval between two consecutive trials. Every time a step (12~15 trials) had been conducted, a five minute interval was inserted.

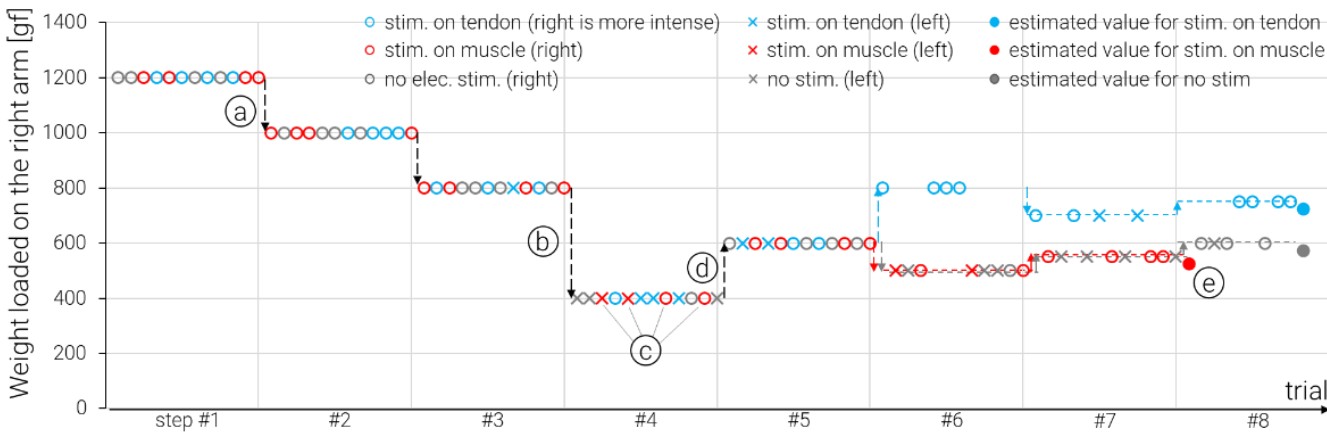

**Figure 2.** Example of the experimental procedure. In each step, the three conditions' trials were conducted randomly. All conditions' steps were simultaneously changed after all the transitions of steps were decided. (**a**) The first step size was 200 gf (rule #1). (**b**) Since the previous two steps moved in the same direction, this step size was be doubled (rule #3). (**c**) For step #4 under the EMS condition, the answers were 2:2, so an additional trial was conducted at the last of this step (rule #2). In this case, he answered as more intense for the left, so the step went up. (**d**) Since the direction was flipped, the step size was halved (rule #3). (**e**) The step size got to 25, and the EMS's condition procedure was stopped (rule #4).

### 3.3. Electrical Stimulator, Stimulation Waveform, and Electrodes' Position

The electrical stimulator [43] that is the same as previously reported [13] was used. We used disposable gel electrodes with a 1.5 cm × 5 cm cut out of 5 cm × 5 cm (Figure 1a). One pair of electrodes was placed over the musculotendinous junction of the biceps on the left arm, and the other pair was placed on the biceps muscle belly. For the electrode pair on the tendon, the GND was on the radial side, and the stimulating electrode was on the ulnar side. For the one on the muscle belly, the stimulating electrode's position on the muscle was adjusted for each participant so that the muscle contraction could be generated efficiently.

The electrical stimulation can control the current intensity up to 15 mA. The current waveform is shown in Figure 3a. The waveform is such that the sum of the charges becomes zero in one pulse, and a large positive current is given to the stimulating electrode. This waveform was used because it was considered to be the least likely to produce unpleasant skin sensations and the easiest to produce force sensations, comparing it with the case where a negative current was given and the bipolar stimulation where the positive and negative current values were the same in the pilot study. The pulse was determined by two parameters: pulse height $H$ [mA] and pulse width $W$ [µs]. The frequency was fixed at 200 Hz.

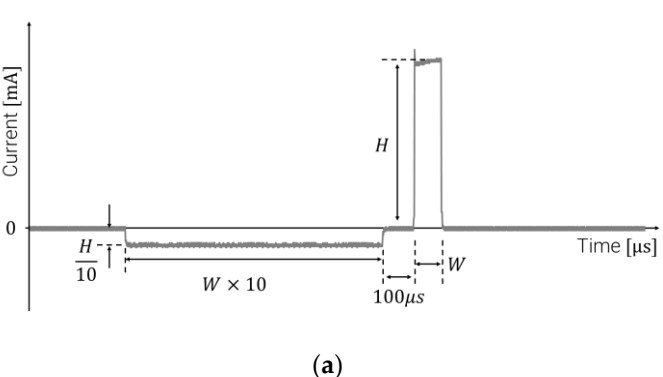
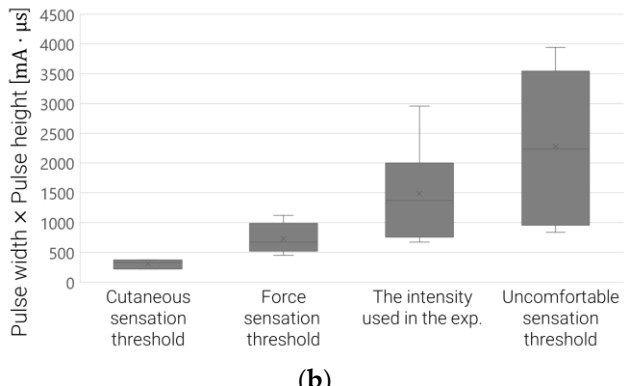

(**a**)

(**b**)

**Figure 3.** (**a**) Electrical stimulation's waveform. (**b**) Electrical current pulse intensity, whose value is equal to $W \times H$ [mA · μs].

For each participant, the current parameters are calibrated as follows. At the beginning of the experiment, the pulse width was set at 100 μs . Then, the current value was gradually increased to measure the cutaneous sensation threshold and the force sensation threshold using the ascending series-only method of limits. In this stage, we confirmed if the participants could feel the force sensation. As a result, all of them could feel the sensation. Next, the pulse height at the force threshold was fixed, the pulse width was gradually increased, and the uncomfortable threshold was measured by the ascending series-only method of limits. Then, we used the force sensation threshold $W_f$ and the uncomfortable threshold $W_p$, determining $(W_f + W_p)/2$ as the current parameter of the electrical stimulation of the tendon. Figure 3b shows the thresholds measured for each participant and the current intensity used in the experiment. The voltage depends on the skin resistance, limited to 300 V as the maximum output. For the electrode pair on the muscle, the same current value as the pulse height of the electrode pair on the tendon was used, and the pulse width was adjusted so that the elbow joint flexed steadily and the posture of the forearm was almost vertical when the force of 400 gf was applied in the extension direction using the device shown in Figure 1b,c.

### 3.4. Participants in the Experiment

Prior to the experiment, participants were briefed on the experiment and asked to sign an informed consent. The participants were eight males between the ages of 21 and 26, six of whom were right-handed, and two reported being left-handed. Before measuring the magnitude of the force sensation, we adjusted the current parameters to see if the electrical stimulation of the tendon produced a force sensation that caused the elbow to open. Since all the participants confirmed that this force sensation was generated, we carried out the experiment for all the participants. In all participants, we confirmed that the stimulation of the muscle belly caused contraction of the biceps brachii muscle and flexion of the elbow joint.

### 3.5. Analysis Method

Since it was necessary to compare each of the three groups of force magnitude data obtained in the experiment, multiple comparisons using the Bonferroni method were conducted. The significance level was set at 5%.

### 4. Results

The experimental results are shown in Figure 4. The bar graph shows the mean value, and the error bars show the standard error. The scatter plot shows the data of each participant. In the muscle condition, one participant's data was obviously different from the other participants' data, i.e., it was much larger than the data in the condition without electrical stimulation, so we labeled this as an outlier. In the Bonferroni method, a paired two-tailed t-test was performed between each group then followed by Bonferroni

correction, so participants with outliers in the muscle condition were excluded from the other conditions, and the groups were compared as $n = 7$. The results showed that the stimulation condition on the tendon was significantly greater than the other two conditions (vs. no-stim.: $p = 0.012$, vs. muscle $p = 0.014$). In contrast, the stimulation of the muscle belly condition was significantly smaller than the one on the tendon but not significantly different from the no-stim condition ($p = 0.28$).

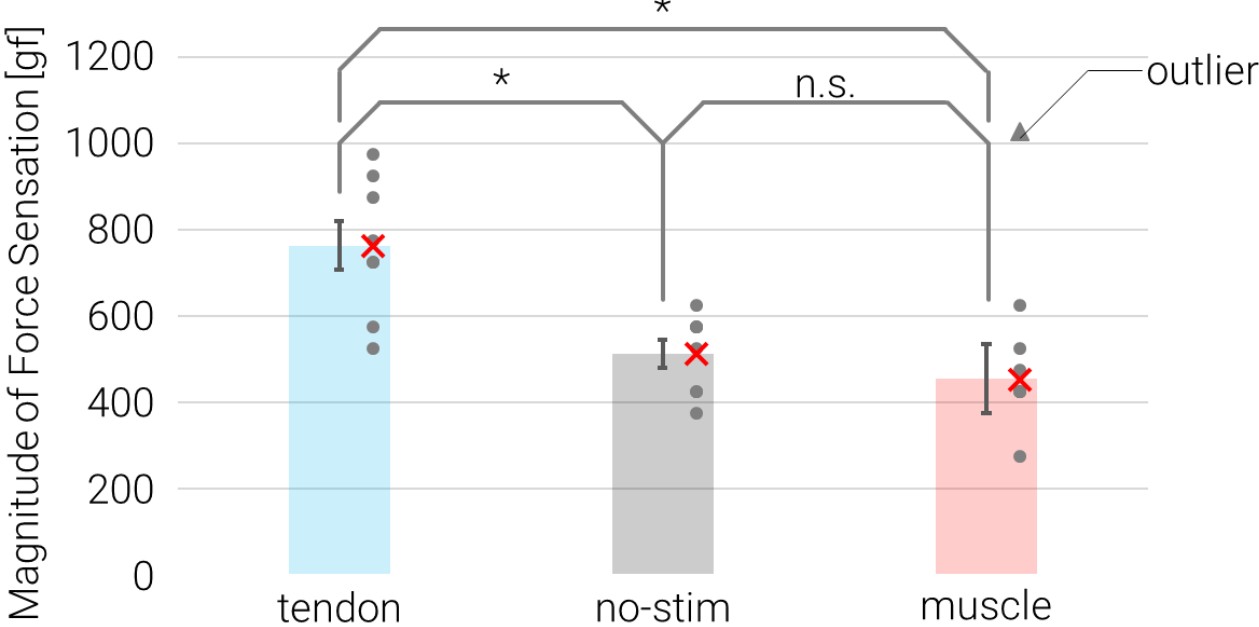

**Figure 4.** Bar plots of the magnitude of force sensation under the three conditions. The scatter plots show each participant's data, some of which are showed overlapped each other. A triangle point in the scatter plot means the outlier. The error bar means standard error. A single star (*) in the figure means significance at 5% level, and the n.s. means no significance.

In the case of electrical stimulation of the tendon, we collected free answers from the participants about how they felt the sense of force, the range of tactile sensation, and how they evaluated the magnitude of the sense of force. All the participants answered that the electrical stimulation of the tendon caused a sensation as if the forearm was pushed forward around the elbow joint, i.e., in the extension direction. In addition, at least three participants reported that when they moved their arms during the force comparison, the stimulation gave them a sensation as if their arms moved more than they expected.

Two participants felt a wide range of the cutaneous sensation around the electrode positions and on the whole forearm, including the hand, during the stimulation. One participant felt stimulation of the forearm, and the cutaneous sensation was felt running along the line from the elbow to the thumb rather than on the ventral surface of the forearm, as the former two mentioned. Two others reported that the cutaneous sensation was limited to the proximal side of the forearm around the elbow from the electrodes' position on the upper arm. The other three reported that the cutaneous sensation was limited just to the area around the electrode on the upper arm, and that they felt a slight tactile sensation on the forearm due to the electrical stimulation.

In the comparison of the magnitude of the force sensation, at least three participants answered that when the weight on the right side was obviously larger, they compared the magnitude of the force from the pressure sensation given by the band to the arm. On the other hand, two participants reported that the cutaneous sensation on the forearm caused by the electrical stimulation masked the pressure sensation on the left arm caused by the band, making it impossible to compare using skin sensation as a cue. In the situation where the skin sensation was not used as a cue, half of the participants reported that they

focused on the displacement and velocity of the forearm movement in the direction of the elbow extending when they moved their arms. None of the participants reported using the difficulty in moving the elbow joint in the direction of flexion as a cue.

Participants were asked how they felt the force sensation and how they evaluated the magnitude of the force sensation in the presence of electrical stimulation of the muscle belly. Three participants, including one who has the outlier (outlier participant), said that the contraction made their arms feel heavier and more difficult to move when they moved their arms. However, one of the participants had a smaller estimated force sensation during electrical muscle stimulation than without electrical stimulation. On the other hand, the outlier participant and the other participant had force estimates that were equal to or at least greater than the stimulation of the tendon and the condition without electrical stimulation. The other participants reported that the EMS canceled out the force exerted by the weights, so that the force sensation felt lighter or the force sensation was entirely lost. Two of them reported that the pressure sensation by the wristband became stronger when the EMS was present, but at the same time, it was not enough to be used for judging the influence of force sensation.

## 5. Discussion

Our hypothesis was that force perception would be (1) increased by stimulation of the tendon and (2) decreased by stimulation over the muscle belly. The experimental results clearly showed that stimulation of the biceps tendon enhanced force perception in the direction of elbow extension. Some people felt that the force sensation was more than doubled. However, there was a large variation, and some participants felt only about a 100 gf ($\simeq 0.98$ N) increase. Even so, the force sensation was enhanced in all eight participants, so we found that our hypothesis (1) is correct. On the other hand, the latter hypothesis (2) was not confirmed.

In this experiment, it took one and a half to two and a half hours to converge the estimates of the magnitude of the force sensation, and we had several break times. So, several participants lost the force sensation induced by the electrical stimulation during the test. In such cases, we adjusted the electrode positions and current parameters so that the participants sufficiently felt the sensation again. In addition, the participants' muscular sizes differed significantly. These factors could have caused the variation in the magnitude of force sensation, and the long time taken to converge the psychophysical estimation may have reduced the accuracy of the values, so a quicker procedure should be adopted in the future.

The stimulation condition on the muscle belly did not significantly reduce force sensation, but several participants reported a sensation of being assisted in supporting the weight. These sensations were significantly different from those induced by the stimulation of the tendon, suggesting that force perception induced by the two electrical stimulation methods are different phenomena. One participant mentioned that electrical muscle stimulation caused him to feel as if he was exerting force himself, which may have been due to the excitation of Golgi tendon organs by muscle contraction, possibly similar to the electrical stimulation of tendons. However, the magnitude of the force sensation for that participant was smaller in the EMS on the muscle belly than in the no-stimulation condition, like almost all the other participants. These are different from the phenomenon induced by stimulation of the tendon, which in contrast did not induce an apparent muscle contraction even when the current was strong as it made an uncomfortable sensation and increased the force sensation, even when the current was much weaker than the uncomfortable intensity. Therefore, we consider a mechanism, which did not relate to muscle contraction, contributed to the force sensation incrementation, e.g., stimulation of the afferent of the Golgi tendon organs.

On the other hand, we cannot reject the explanation that the muscle contraction contributes to the force sensation, since the outlier participant, who answered that the force sensation increased even when EMS was applied through the electrodes above the muscle

belly, reported that it was due to the sensation that the muscle stimulation made it tough to move their arm. In addition, another participant reported that while the sensation on the right arm (receiving only the real force by the weight) could be described as "moving the arm in the water", the left arm with EMS could be described as "moving the arm in the mud". The outlier participant reported that the muscle contractions became less constant, and their arms began to vibrate involuntarily when he moved his arm. Since the biceps belly moves extensively as the elbow bends, we considered that the arrangement of the muscle and electrodes was changed by the EMS, resulting in phasic movement depending on the angle of the arms.

The no-stim condition's mean was also compared to the reference force (400 gf) by conducting the t-test, the significance level of which was set at 5%, and it was significantly larger than 400 gf ($p = 0.010$). Only a left-handed participant slightly underestimated the magnitude of no-stim, while the other left-handed overestimated it more than some right-handed participants. On the other hand, one right-handed participant, who reported he was a tennis player and his right arm's muscle was larger than the left, scored the sensation magnitude under the no-stim (as well as under the other conditions) slightly larger than the mean, that is, he had a typical score. Therefore, we could not confirm that handedness has a dominant effect; indeed, there were too few participants to conclude that.

In future work, we have to conduct some multimodal experiments to confirm if this proposed method is useful in an applied environment such as VR or remote communications. This study has just shown that stimulation of the tendon can present force sensation without physical movement and is still limited to the basic research. Additionally, in order to use this proposed method adequately as a way to present one single sense modality, we have to reveal the mechanism of this in detail.

The tactile sensation path can be considered as another candidate that could induce the force sensation, other than the Golgi tendon organ. In the condition of electrical stimulation of the tendon, the tactile sensation tended to occur in the forearm because nerve bundles reside in a relatively shallow area where the electrodes were placed. Even so, some participants felt no tactile sensation on the forearm, but they felt a force sensation as if the elbow was stretched. This suggests that the contribution of deep sensation was greater than that of cutaneous sensation. Indeed, since some studies have shown that cutaneous sensory nerves contribute to proprioception [44], we cannot rule out the possibility that the cutaneous sensory nerves on the ventral side of the elbow were stimulated and processed centrally only as force sensation without cutaneous vibration or pressure sensation. However, some participants reported that the force illusion did not occur when their posture changed largely from the initial posture, suggesting that the stimulation of deep sensory nerves, which are easily affected by posture changes, was the main factor.

The muscle spindles' contribution to the force sensation should be considered. However, we estimate its contribution as small because none of the participants reported kinesthetic illusion like that induced when the vibration stimulation is applied. On the other hand, some participants felt they moved their arms more or faster when they voluntarily moved their arms while the electrical stimulation was applied to the tendon. This could be interpreted as a motion illusion caused by muscle spindles, but it could also be interpreted as the participants recalling acceleration from the perceived external force. Indeed, it has been reported that electrical stimulation of the tendons could produce a phenomenon similar to the motion illusion [45].

We observed another interesting phenomenon. At least three participants reported that they experienced involuntary extension movements of the elbow joint at the moment of electrical stimulation of the tendon of the flexor. If it had worked as EMS, the elbow would have moved in a flexing direction rather than the extension. This might be explained by the Ib reflex, which is regarded to be contributed to by the Golgi tendon organ. This reflex suppresses the excitation of motor nerves to protect the muscle when a massive force is applied to the muscle instantaneously. This inhibitory reflex has been confirmed to

occur when needle electrodes stimulate the Golgi tendon organ or Ib nerve. So, in order to elucidate the mechanism of force illusion induced by electrical stimulation of tendons, measuring these reflexes with the setup of the present experiment should be conducted as future work.

## 6. Conclusions

In this paper, we quantitatively evaluated the presence, direction, and amount of force sensation induced by electrical stimulation of the tendon (musculotendinous junction) and on the muscle belly regions of the biceps brachii muscle. The results showed that the stimulation of the tendon produced significant force sensation incrementation in the direction of elbow extension. On the other hand, the stimulation of the muscle belly assisted the weight to support the external force, resulting in the weakening of force sensation in some participants; one participant felt that they were exerting themselves strongly, and others answered that the sensation of force was enhanced because the muscle was jammed to move. In general, we can conclude that the sensations produced by muscle stimulation and tendon stimulation are quite different. Although the contribution of Golgi tendon organs to the force sensation can be explained by multiple observations, we cannot conclude this at this stage and need to focus on the reflexes specific to each receptor and nerve to elucidate the mechanism.

**Author Contributions:** Conceptualization, all authors; methodology, all authors; validation, A.T..; investigation, A.T.; resources, all authors; data curation, A.T.; writing-original draft preparation, A.T.; writing-review and editing, H.K.; visualization, A.T.; supervision, H.K.; project administration, H.K.; funding acquisition, all authors. Both authors have read and agreed to the published version of the manuscript.

**Funding:** This research was funded by JSPS KAKENHI Grant Number JP20J14306, JP18H04110.

**Institutional Review Board Statement:** The study was conducted according to the guidelines of the Declaration of Helsinki and approved by the Institutional Review Board of The University of Electro-Communications (#18044).

**Informed Consent Statement:** Informed consent was obtained from all participants involved in the study.

**Data Availability Statement:** Data sharing not applicable.

**Conflicts of Interest:** The authors declare no conflict of interest.

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
