# Peer review of "Force Sensation Induced by Electrical Stimulation of the Tendon of Biceps Muscle"

_applsci, doi:10.3390/app11178225_

Round 1
Reviewer 1 Report
In this study, the authors investigated the artificial sensation of the force when the tendon or the muscle was stimulated. An experiment that subject carrying the different weight by the left and right arms were performed, and weight was modified to find the force sensation produced by the stimulation. This is very interesting and I’m excited to see this technology in the future VR game! Several comments are below:
- The stimulation is on the tendon bicep and muscle belly. But the stimulation is on the skin surface. The electronic signal can go to both the tendon and muscle. how can you separate the effects through tendon and muscle? From Fig.1a, it seems that two stickers represent tendon and muscle. But the length of tendon and muscle vary among people. How can you make sure the tendon sticker electrical signal just goes to the tendon. (If someone tendon is short, the signal from the tendon sticker may go to the muscle belly)
- What is gf?
- The strength of left and right arms are different. If one guy has a super strong right arm and weak left arm, will this make difference?
- How long is the experiment? When people lift weights, they lose strength when they lift them for a long time. So probably they will feel heavier after a long time. Will this be a problem?
Author Response
Response to Review 1 comments:
- The stimulation is on the tendon bicep and muscle belly. But the stimulation is on the skin surface. The electronic signal can go to both the tendon and muscle. how can you separate the effects through tendon and muscle? From Fig.1a, it seems that two stickers represent tendon and muscle. But the length of tendon and muscle vary among people. How can you make sure the tendon sticker electrical signal just goes to the tendon. (If someone tendon is short, the signal from the tendon sticker may go to the muscle belly)
 Thank you for your insightful question. In the present study, we did not deny that on-tendon conditions stimulate the Golgi tendon organ by indirectly stretching the tendon through a small contraction of the muscle. In fact, some subjects felt an increase in force sensation in the on-muscle condition similar to that in the on-tendon condition. In the on-tendon stimulation condition, however, when we investigated the discomfort threshold (pain threshold) as the upper limit of the current to be used, all participants did not experience at least such muscle contraction as making arm flexion at the discomfort threshold. Also, the force sensation was enhanced at the electrical stimulation intensity weaker than the discomfort threshold in the on-tendon condition. We believe this may be due to a non-muscle stimulation pathway, i.e., the stimulation of sensory nerves extending from the Golgi tendon organ. We have added a description of this in the Discussion section (l.387-395).
- What is gf?
Thank you for your question. The [gf] represents gram force, where . An explanation of this has been added in l.65-66. The value in [N] is also added.
- The strength of left and right arms are different. If one guy has a super strong right arm and weak left arm, will this make difference?
Thank you for your sharp remarks. Among the participants, one right-handed person with tennis experience self-reported having large muscles in his right arm, but the results were similar to the overall trend. Merging it with points made by other reviewers, we have added a description of this in the Discussion section (l.406-414).
- How long is the experiment? When people lift weights, they lose strength when they lift them for a long time. So probably they will feel heavier after a long time. Will this be a problem?
The maximum experimental time per participant was 2 hours and 30 minutes, and although we did not measure the time per trial, it was not more than 2 minutes at most. Ten to thirty seconds were left between trials due to the exchange of weights. In addition, there was a break of about 5 minutes every step (every 12 to 15 trials). We have added a description of this in section 3.2. (l.244-247)

Reviewer 2 Report
Overall:
Overall this work is interesting to the readers and in line with the shift in the tactile feedback interfaces. The authors tried to investigate if the force sensation is felt by an individual by electrically stimulating at the tendon or at the muscle belly, by conducting a human-subject study under three conditions. The experiment appears to be well designed. However, there are a few questions and suggestions to improve clarity.
Introduction:
Line 38-39: The authors mentioned, "However, the multiple degrees of freedom of motion require the same number of motors, causing an increase in the size of the device in order to have multiple degrees of freedom". There are a few papers that talk about the successful tactile feedback transmission with dimensional reduction, such as,
N. Landin, J. M. Romano, W. McMahan and K. J. Kuchenbecker, "Dimensional reduction of high-frequency accelerations for haptic rendering", Haptics: Generating and Perceiving Tangible Sensations (EuroHaptics), pp. 79-86, 2010.
I was wondering if the authors would want to state the pros of using electrical stimulation if it can solve with only 1 motor based actuator.
Materials and Methods:
Line 169: How did you select the exact location of Golgi tendon organs? Adding this information can improve clarity.
Some lines in the experimental procedure need clarity, such as :
Line 222-223: "We ran all the estimation procedures for all the three conditions in parallel, and each trial of the three conditions was conducted randomly". What is exactly meant by parallel? Does it only mean three conditions in the same step?
Line 225: Were the participants given any physical limitations for knowing the range of movements they make? How do the participants know if they move only 10 degrees to 20 degrees?
Line 240-241: Did the authors try to see the effect of handedness? Why did the authors stimulate only the left hand?
Line 244: The authors stated, "the stimulating electrode’s position on the muscle was adjusted for each participant so that the muscle contraction could be generated efficiently." Was this statement based on some quantitative or qualitative measurements?
Discussion:
Although the authors tried to explain the possibility of not confirming the 2nd hypothesis with the duration of the experiment, they did not explain the fact why is there any significant difference between stimulation on muscle belly and no stimulation condition?
General:
-- No information on ethical approval for the study is added.
Do the authors have any plans for a similar experiment in multimodal stimulation and see the effect?
Author Response
Please see the attachment.
Response to Review 2 comments:
Introduction:
Line 38-39: The authors mentioned, "However, the multiple degrees of freedom of motion require the same number of motors, causing an increase in the size of the device in order to have multiple degrees of freedom". There are a few papers that talk about the successful tactile feedback transmission with dimensional reduction, such as,
- Landin, J. M. Romano, W. McMahan and K. J. Kuchenbecker, "Dimensional reduction of high-frequency accelerations for haptic rendering", Haptics: Generating and Perceiving Tangible Sensations (EuroHaptics), pp. 79-86, 2010.
I was wondering if the authors would want to state the pros of using electrical stimulation if it can solve with only 1 motor based actuator.
Thank you for the important reference. I should have mentioned that there are several studies that cover multiple degrees of freedom with a single actuator. Even though one actuator can cover multiple axes, the situations are limited, and the device becomes heavy as you increase the output even if it has only one motor. EMS does not use a single motor, and electrical stimulation of sensory nerves can excite the nerves without regard to the mechanical structure of the receptors. Because of these features, we believe that electrical stimulation can achieve this with a small footprint, especially when considering the presentation of large force sensations. However, stimulation from surface electrodes can cause unpleasant skin sensations as well as precise force control. We have added a description of this in the Introduction chapter. (l.40~43, 61~63)
Materials and Methods:
Line 169: How did you select the exact location of Golgi tendon organs? Adding this information can improve clarity.
Thank you for pointing this out. In our study, we considered that the electrical stimulation on the tendon stimulated the Golgi tendon organ, but we were not able to determine this. However, we placed the electrodes on the area where the muscle tapers away from the belly and connects to the tendon (myotendinous junction). At this location, no muscle contraction occurred even when the electrical stimulation was uncomfortably loud. The absence of muscle contraction and the location of the electrodes near the myotendinous junction suggested that stimulation of the Golgi tendon organ was a plausible explanation for the force sensation. In order to confirm the position of the muscle belly and tendon, the biceps muscle was contracted by applying a certain amount of force in the same direction as the pull of the weight while the participant kept the same posture as during the experiment, which brought up the muscle belly and tendon and we confirmed them visually. A description of this was added to Materials and Methods. (l.181~186)
Some lines in the experimental procedure need clarity, such as :
Line 222-223: "We ran all the estimation procedures for all the three conditions in parallel, and each trial of the three conditions was conducted randomly". What is exactly meant by parallel? Does it only mean three conditions in the same step?
Thank you for your question. In this case, the experiment proceeded, as shown in Fig.2. In the Materials and Methods, we tried to clarify that the explanation and that Fig.2 are related. (l.240)
Line 225: Were the participants given any physical limitations for knowing the range of movements they make? How do the participants know if they move only 10 degrees to 20 degrees?
Thank you for pointing this out. This was just a verbal instruction that they could move their arms within the range of 10 to 20 degrees, and we did not measure or give any feedback to see if it was actually within this range. I have added a sentence to clarify this. (l.243-244)
Line 240-241: Did the authors try to see the effect of handedness? Why did the authors stimulate only the left hand?
Thank you for your sharp remarks. I used only the right hand for the simplicity of the experiment, and I thought that if there was a difference in handedness, it would show up as a cluster in the data. Of the two left-handed participants, only one underestimated the weight in the no-stimulation condition, while the other overestimated it more than some of the right-handed participants. Due to the small number of participants, no statistical conclusions can be drawn, but within the scope of this experiment, we could not confirm the effect of handedness. We have added a description of this in the Discussion section, along with the points raised by another reviewer. (l.406-414)
Line 244: The authors stated, "the stimulating electrode's position on the muscle was adjusted for each participant so that the muscle contraction could be generated efficiently." Was this statement based on some quantitative or qualitative measurements?
Thank you for pointing out the shortcomings. The determination of the position of the electrodes was done mostly heuristically. However, the position of the muscle belly and tendon of the biceps muscle was visually confirmed by applying a certain amount of force in the direction of that induced by weight and contracting the biceps muscle in the posture of the experiment. In addition, since the effect of muscle contraction can be dramatically changed by moving the electrode by less than 1 cm, we believe that the heuristic method has a certain degree of persuasiveness. We have added a description of this in the Materials and Methods chapter. (l.181~186, 200~203)
Discussion:
Although the authors tried to explain the possibility of not confirming the 2nd hypothesis with the duration of the experiment, they did not explain the fact why is there any significant difference between stimulation on muscle belly and no stimulation condition?
Thank you for your question. N.S. in Fig.4 stands for No Significance. In this experiment, there was actually no significant difference between supramuscular stimulation and no stimulation conditions. The caption of Fig. 4 has been modified to make this clearer. (l.318-320)
General:
-- No information on ethical approval for the study is added.
Thank you for pointing this out. We had described the approval by the ethics committee in the Institutional Review Board Statement as the data of the paper, but we did not describe it in the text. We have added a description of this in Materials and Methods. (l.172~174)
Do the authors have any plans for a similar experiment in multimodal stimulation and see the effect?
Thank you for your question. Yes, we do. We have conducted multimodal experiments on the combination of electrical stimulation of the dorsal wrist, vibration (collision) presentation to the dorsal hand, and visual presentation. We will continue to conduct multimodal experiments using electrical stimulation of the biceps tendon to confirm its practicality in the VR space. We added to the Discussion chapter. (l. 415-418)

Round 2
Reviewer 1 Report
nice work!